# Protective Effect of Alkaline Phosphatase Supplementation on Infant Health

**DOI:** 10.3390/foods11091212

**Published:** 2022-04-21

**Authors:** Haoming Wu, Yang Wang, Huiying Li, Lu Meng, Nan Zheng, Jiaqi Wang

**Affiliations:** 1State Key Laboratory of Animal Nutrition, Institute of Animal Sciences, Chinese Academy of Agricultural Sciences, Beijing 100193, China; wuhaoming@caas.cn (H.W.); lihuiying@caas.cn (H.L.); menglu@caas.cn (L.M.); wangjiaqi@caas.cn (J.W.); 2Laboratory of Quality and Safety Risk Assessment for Dairy Products of Ministry of Agriculture and Rural Affairs, Institute of Animal Sciences, Chinese Academy of Agricultural Sciences, Beijing 100193, China; 3Key Laboratory of Quality & Safety Control for Milk and Dairy Products of Ministry of Agriculture and Rural Affairs, Institute of Animal Sciences, Chinese Academy of Agricultural Sciences, Beijing 100193, China; 4State Key Laboratory of Membrane Biology, Tsinghua University-Peking University Joint Center for Life Sciences, School of Life Sciences, Tsinghua University, Beijing 100084, China; wangyang881229@mail.tsinghua.edu.cn

**Keywords:** ALP, infant intestinal health, raw milk, inflammation, allergy

## Abstract

Alkaline phosphatase (ALP) is abundant in raw milk. Because of its high heat resistance, ALP negative is used as an indicator of successful sterilization. However, pasteurized milk loses its immune protection against allergy. Clinically, ALP is also used as an indicator of organ diseases. When the activity of ALP in blood increases, it is considered that diseases occur in viscera and organs. Oral administration or injecting ALP will not cause harm to the body and has a variety of probiotic effects. For infants with low immunity, ALP intake is a good prebiotic for protecting the infant’s intestine from potential pathogenic bacteria. In addition, ALP has a variety of probiotic effects for any age group, including prevention and treatment intestinal diseases, allergies, hepatitis, acute kidney injury (AKI), diabetes, and even the prevention of aging. The prebiotic effects of alkaline phosphatase on the health of infants and consumers and the content of ALP in different mammalian raw milk are summarized. The review calls on consumers and manufacturers to pay more attention to ALP, especially for infants with incomplete immune development. ALP supplementation is conducive to the healthy growth of infants.

## 1. Introduction

Milk and dairy products may contain a variety of microorganisms, which may be an important source of foodborne diseases [1,2]. To eliminate pathogenic microorganisms in milk, processing plants choose to use the heating method for sterilization. Negative ALP activity is used to confirm the successful pasteurization of skimmed or whole milk [3]. Compared with most pathogenic bacteria, ALP has slightly higher heat resistance (71.6 °C for 15 s). Therefore, ALP activity is used to measure the pasteurization degree of beverages, especially milk and dairy products [3,4,5]. Pasteurization is the standard method for eliminating pathogens. Inadequate or defective pasteurization does not kill all foodborne pathogens [5]. The ALP activity in raw milk varies with the source of raw milk. China, the United States, and European countries require that the ALP activity in pasteurized beverages be less than 350 mU/L [3,6].

However, negative ALP activity results in a loss of inhibition activity against the toxicity of Gram-negative bacteria LPS in milk [7]. ALP can remove the phosphate bond on LPS and then remove the pathogenicity of LPS. After sterilization, LPS existing on bacterial cell membranes in raw milk will be released into milk. However, ALP-negative milk will no longer have the dephosphorylation of LPS. LPS with inflammatory activity in dairy products may pose a threat to consumers [7]. At the same time, some studies pointed out that pasteurized milk lost its protective effect on food allergy and that its protective ability recovered after adding ALP [8].

Alkaline phosphatase (ALP) is an enzyme that catalyzes the hydrolysis of phosphate under alkaline conditions. ALP widely exists in various mammalian tissues and plays an important role in biological processes. However, human tissues can also synthesize ALP, including tissue nonspecific alkaline phosphatase (TNAP), placental alkaline phosphatase (PLAP), germ cell alkaline phosphatase (GCALP), and intestinal alkaline phosphatase (IAP) [9]. However, for infants with incomplete immune development, ALP gene expression is still low, and exogenous ALP supplementation is very necessary [10]. At the same time, ALP protects the health and stability of tissues and bodies at all life stages. When diseases occur, ALP is used as an important biomarker for the diagnosis of many diseases [11,12,13].

In this paper, the prebiotic effects of ALP on the infant intestinal tract and allergy prevention are summarized; ALP, whether oral or injection, will not pose a threat to the body’s health and can treat or prevent the body’s diseases. At the same time, the content of ALP in different mammalian raw milk that was determined in different studies is statistically summarized. Through the introduction of this paper, we hope to appeal to food processors and consumers to pay more attention to ALP and to provide theoretical support for further improving the level of nutrition and health.

## 2. Important Health Indicators

According to clinical diagnosis, abnormal ALP activity in blood is related to various diseases—for example, bone-specific ALP and bone disease [11], breast cancer diagnosis [12], and diabetes [13]. In the case of diagnosis, ALP measured in serum is used as a diagnostic tool for liver disease [14] and testicular cancer [15,16]. In patients with chronic liver disease, blood ALP increased in the order of chronic hepatitis (CH), liver cirrhosis (LC), and hepatocellular carcinoma (HCC) with pathological progress [17]. At the same time, ALP in the patient’s blood will be detected before and after a medical process. The change in ALP in blood can affect the therapeutic effect and survival rate of patients, such as metastatic prostate cancer [18], metastatic breast cancer [19], clear cell chondrosarcoma [20], and metastatic nasopharyngeal carcinoma [21].

## 3. Factors Affecting ALP Activity

IAP has an important protective effect on the infant intestine. IAP is a homodimer, and each subunit consists of two Zn^2+^ and one Mg^2+^ ion [22]. The activity of IAP was the highest at pH 9.7 [23]. IAP is generally secreted in the duodenum, followed by less expression in the jejunum, ileum, and colon [24,25,26]. IAP is largely absent in the stomach in an acidic environment [26,27]. IAP mainly exists in lumen vesicles secreted by intestinal cells on the brush edge of microvilli. At the same time, a small amount of IAP will also be bidirectionally released into the blood and lumen and then spread throughout the intestine [28]. The investigation found that the level of IAP activity in human intestinal tract varies with blood type. The level of IAP in human intestinal tract of type O and type B is the highest, while the activity in human intestinal tract of type A blood is the lowest [29]. In the gut, IAP is a mucosal defense factor that restricts bacteria from crossing the mucosal barrier into mesenteric lymph nodes [30,31]. In addition to the local activity of IAP in intestinal mucosa, about 1–2% of IAP is released into the blood or gastrointestinal cavity to treat and prevent systemic infection and sepsis [32].

IAP activity is related to dietary nutrition and dietary frequency. Food intake can regulate the activity of IAP. Intake of omega-3 fatty acids can reduce the level of LPS and improve intestinal permeability by increasing the activity of intestinal IAP in transgenic mice [33]. In addition, a protein diet may reduce the activity of IAP. When feeding calves before rumination with a soybean diet, the activities of intestinal enzymes such as IAP in the intestine are reduced [34]. Another study found that the intestinal IAP activity of pigs increased significantly in the short term after using wheat or barley instead of a milk-based high protein diet [35]. After the protein structure of milk is destroyed by fermentation, intake of yogurt can increase the activity of IAP [36]. In addition, glucomannan, oligosaccharide, and vitamin D supplementation are associated with increased intestinal IAP activity [37,38,39]. Some phytochemical components, such as curcumin, black pepper, red pepper, ginger, piperine, and capsaicin, have also been found to be associated with increased IAP activity [40]. When mice were fasted for two days, IAP expression in the intestine decreased significantly, resulting in a decrease in the ability of intestinal LPS dephosphorylation [31]. Because starvation leads to the downregulation of IAP, the host’s sensitivity to pathogens increases. Therefore, for the treatment and prognosis stage of patients, intake of diet that can increase IAP is of positive significance for the recovery of patients’ health [31] because IAP is associated with limiting the speed of fatty acid transmembrane transport to intestinal cells [41,42]. HFD feeding increases IAP stress-induced secretion in the intestine, thereby maintaining host weight stability [43]. On the other hand, some studies have found that HFD feeding can reduce IAP activity and increase TLR4 activity in the intestine of obese rats [44]. Similarly, when present in food ω- 3 PUFA will also lead to a decrease in IAP expression and activity [45]. IAP-KO mice fed HFD were more likely to gain weight [41]. Therefore, for different individuals, the effect of HFD feeding on intestinal IAP is not stable, and further research is needed. In addition, protein intake may also be related to the regulation of IAP. The activity of ALP in the intestine of rats fed a protein-free diet decreased by 36–38% [46]. In conclusion, the intake of food nutrition is related to the activity of ALP. Reasonable dietary intake is helpful for regulating the activity of intestinal ALP.

## 4. Infant Intestinal Health

In Table 1, the research papers on the prevention and treatment of diseases by ALP are summarized. Relevant studies were identified by searching Web Science, Google Scholar, and PubMed. If a study met the following criteria, it was included in the table: 1. Study and evaluate the effect of oral or injection ALP treatment; 2. Research papers written in English until December 2021. All studies that met the requirements of direct ALP treatment of laboratory animals are listed in the table. It turns out that high IAP activity in the intestine of full-term newborns, coupled with the high ALP activity in breast milk in the first few days after birth, provides sufficient detoxification capacity for LPS of initially colonized bacteria [10]. After the onset of IBD in infants, TLR4 mRNA expression and protein levels in inflammatory colonic mucosa in children increased [47]. The increased expression of TLR4 may be related to the content of IAP in intestinal mucosa. IAP activity below normal level may lead to IAP/TLR4 imbalance, resulting in increased sensitivity of mucosa to LPS [25]. After treatment, the recovery of intestinal mucosa is very important for the prognosis of the disease. Therapeutic treatment to restore intestinal flora balance may have a significant impact on mucosal healing of IBD [48]. In an animal model of colitis induced by sodium dextran sulfate (DSS), exogenous administration of IAP improved the symptoms of colitis [49]. Compared with wild-type mice, IAP-KO mice had more severe colitis induced by DSS [32]. In the case of severe intestinal epithelial injury, oral IAP may have beneficial effects [50]. The results obtained in the pediatric population also demonstrate that oral IAP may be beneficial for children with IBD [25].

Early in life, the initial colonization of intestinal microorganisms will affect the development of intestinal host defense [51,52], and the appropriate development of intestinal tissue may have a far-reaching impact on immune health in infancy and even throughout life [53,54]. Because the intestinal immune tissue is immature at birth and develops with the initially colonized microorganisms, it has been suggested that the increase in allergy and autoimmune diseases may be caused by the interference of microbial colonization and development of intestinal host defense system [55]. It is worth noting that the host defense capacity of the intestine of preterm infants is particularly immature and that the initial colonization of intestinal microorganisms is not the same as that of normal-born infants [56]. Compared with mature human intestinal cells, infant intestinal cells are more likely to produce excessive inflammatory response when stimulated and even respond to some intestinal symbiotic bacteria with an inflammatory effect [57,58]. The imbalance in intestinal microbial ecology in the neonatal period of preterm infants may lead to excessive intestinal inflammation, resulting in NEC. As the sole source of nutrition for infants, breast milk helps healthy bacterial colonization in the infant’s intestines [59]. Studies have pointed out that breast milk extruded by preterm mothers can protect the infant’s intestine from NEC [60,61]. Although breast milk contains many immune nutrients, such as IgA, oligosaccharides, lactoferrin, ALP, etc., it may help to prevent intestinal inflammation and NEC. At the same time, breast milk may also play an active role in stimulating health-promoting bacteria, thus providing protection against NEC. Studies have pointed out that giving probiotics to preterm infants can also prevent NEC. The combination of probiotics (e.g., Lactobacillus acidophilus and Bifidobacterium bifidum) and human milk has a protective effect, but the combination of probiotics and infant formula has lost its protective effect [62,63].

Compared with infant formula, breast milk has several unique factors to actively protect the intestine of newborns. First, oligosaccharides in breast milk provide an energy source for the intestinal microbiota, help the growth of infant intestinal probiotics, and are beneficial to the infant intestine [64,65]. Second, breast milk itself contains a variety of bacteria that actively colonize the intestine, providing protection for the infant’s primitive microbiota [66]. Third, breast milk contains immunobioactive factors, such as secretory IgA, which can change the colonization of infants’ intestines and protect infants from pathogens [67,68]. Fourth, gastrointestinal administration of exogenous IAP can improve intestinal inflammation and promote intestinal tissue regeneration, while intestinal and systemic IAP administration can reduce systemic inflammation [69]. In addition, oral IAP supplements may regulate intestinal metabolic homeostasis by stimulating intestinal IgA secretion [12]. In conclusion, breast milk not only can change the intestinal environment of infants to prevent pathogenic bacteria but also can promote the colonization of symbiotic bacteria, so as to promote the short-term and long-term immune health of the host.

IAP can maintain intestinal immune balance and improve host tolerance to symbiotic microbiota by reducing lumen ATP concentration and dephosphorylating bacterial LPS [70]. At the same time, IAP knockout mice showed increased fat absorption and obesity, suggesting that intestinal lipid transport is related to the regulation of IAP [36]. Both endogenous and oral IAP supplementation can inhibit the absorption of LPS in dietary fat, and oral IAP supplementation can prevent and reverse metabolic syndrome. In addition, IAP supplementation improved the blood lipid status of mice fed standard low-fat food [36,71]. In animal models of intestinal injury, oral IAP reduces intestinal epithelial injury and inflammation [50]. The dynamic transformation of the form of ALP isozyme is related to the maturation of fetal intestine [72], suggesting that ALP activity may change during human fetal development. Supplementation of ALP in the intestine of newborn rat pups has a protective effect on experimentally induced NEC [73]. ALP has a protective effect on the intestine of premature young rats against intestinal damage and inflammation caused by microbial LPS [74].

Intestinal alkaline phosphatase (IAP) can maintain intestinal health through a variety of mechanisms, including detoxification of lipopolysaccharide (LPS), flagellin, CpG DNA and nucleotides; upregulation of the expression level of tight junction protein, thereby increasing intestinal barrier function; and regulation of intestinal microbiome homeostasis [75,76,77]. During lactation, infant intestinal ALP gene expression and enzyme activity remain low [78]. Preterm birth and formula feeding are considered to be associated with the inhibition in IAP expression and activity, and the lack of ALP may increase the risk of NEC [79]. For infants, breast milk is the only exogenous source of ALP [10], and pasteurization destroys 99% of ALP in breast milk [80]. ALP in breast milk is considered to be an anti-inflammatory factor in the neonatal intestine and a key component in inhibiting NEC [81,82]. The lack of IAP activity will lead to neonatal intestinal ecological imbalance and bacterial translocation, resulting in a variety of diseases [83]. 

In previous studies, it was found that taking LPS can lead to weight gain and acute inflammation and eventually insulin resistance [84]. Similar effects can be seen in animal models fed HFD or LPS. Similarly, LPS can also lead to a pro-inflammatory response in the fetal brain after infecting the mother, thereby increasing anxiety and reducing social activities [85]. Therefore, researchers believe that the important roles of HFD and LPS in the pathogenesis of ASD seem to be consistent between humans and animals. ALP is considered to be a drug that can protect the balance in the intestinal environment and reduce inflammation [86]. ALP has anti-inflammatory effects, prevents intestinal leakage, and promotes a healthy microbiota [70]. At the same time, recent studies have proved that oral supplementation of IAP to pregnant mothers is of positive significance to the health of infants. A mouse model test found that maternal IAP treatment can alleviate some autism spectrum disorder (ASD)-like symptoms of offspring mice [87]. It is worth noting that the content of ALP in breast milk decreased with lactation time [88]. Therefore, the intake of ALP supplementation for infants and mothers is of great significance for maintaining infant health. Unfortunately, ALP does not exist in any infant formula.

## 5. Inhibition of Allergy

Children growing up on farms have a lower risk of asthma and allergies than children living in the same rural area but not on farms [89,90]. This protective “farm effect” is recognized in many people until adulthood [91]. The farm exposure associated with this allergic protective effect appears to be eating unprocessed raw milk [92,93]. In particular, the consumption of raw milk and the protective effect of raw milk have nothing to do with farm conditions, so it can protect the ordinary non-agricultural population [94,95]. Many studies have shown that eating untreated raw milk can prevent the risk of asthma and allergic diseases [92,93,94,95,96]. These epidemiological findings recently confirmed the causal evidence that biological milk can prevent allergic asthma caused by household dust mites and OVA-induced food allergy in mouse models [97,98].

Pasteurization makes milk lose the ability to protect consumers from allergies. The histone acetylation degree of Th1-, Th2-, and regulatory T cell-related genes in splenocyte CD4+ T cells of rats treated with raw milk was higher than that of mice treated with pasteurized milk. Compared with processed milk products, the histone acetylation degree of Th2 gene in rats treated with raw milk was lower. In the study of mice allergic to food, raw milk reduced allergic symptoms to food allergens other than milk. The activation of T cell-related genes is considered to be the cause of the observed tolerance induction, indicating that epigenetic modification helps raw milk protect the body from allergy [98]. On the other hand, some studies have shown that pasteurized milk lost its allergic protection, but pasteurized milk added with ALP restored its allergic protection [8]. Skimmed raw milk inhibited food allergy symptoms similar to raw milk and reduced acute skin allergy and lowered levels of OVA-specific Ig-E- and Th-2-related cytokines. This indicates that the fat component is not an ingredient in raw milk that inhibits food allergy [8].

ALP can regulate the structure of intestinal microbiota and protect consumers from allergy induction. Raw milk treatment increased the relative abundance of butyrate producing bacteria in mouse intestine, in addition to increasing Lachnospiraceae ucg-001, Lachnospiraceae ucg-008, and Ruminiclostridium 5 (Clostridium clusters xiva and IV), and decreased the relative abundance of bacterial Proteus that can promote inflammation. Clostridium clusters xiva and IV can decompose nondigestible oligosaccharides to produce acetic acid, propionic acid, and butyric acid. These organisms are usually close to host epithelial cells, which have a great impact on the host immune balance [99,100]. In a study of mice intestinal microorganisms, Clostridium clusters xiva and IV can help induce the accumulation and differentiation of Foxp3 + Treg cells in mouse colon. Clostridium can induce colon epithelial cells to release active TGF-β and other Treg inducing factors, which can induce Treg differentiation by regulating CD103+ DC. In addition, Clostridium has also been shown to induce IL-10 expressing Treg cells in the colon. Increasing the colonization of intestinal Clostridium can increase the body’s resistance to allergy [101]. These microbes disappeared in the intestines of mice fed pasteurized milk. However, after the addition of ALP to pasteurized milk, the changes in the intestinal probiotic microbiota observed in the raw milk treatment group reappeared and reduced allergic reactions [102].

## 6. Diabetes and Metabolic Syndrome Prevention

The supplement of alkaline phosphatase not only has a prebiotic effect on infants but also has a positive effect on the elderly and even adults with weak immunity. A study of endocrine diseases found that the supplement of ALP may be helpful in preventing type 2 diabetes mellitus (T2DM). T2DM is an important global metabolic disease. Because type 2 diabetes patients have high blood sugar concentration and symptoms that are easily complicated, T2DM has a serious impact on medical expenses, incidence rate, and mortality rate. The hyperglycemia concentration of T2DM is similar to immersing organs in high glucose medium. Long-term illness can lead to long-term injury, dysfunction, and even failure of organs such as eyes, kidneys, nerves, heart, and blood vessels [103,104,105]. From the perspective of etiology, many factors are considered to be related to the occurrence of T2DM, such as autoimmune level, metabolic syndrome, dietary conditions, weight, external infection, genetic information, drug use, stress, pregnancy, etc. [105,106]. Recently, low-grade systemic inflammation caused by continuously elevated levels of endotoxin (LPS) in the blood (metabolic endotoxemia) has been considered to be a cause of T2DM [107]. Studies have shown that high levels of IAP have a protective effect on T2DM patients, whether obese or not. Obese patients with a high level of IAP (about 65 U/g feces) generally do not develop T2DM. When the activity of ALP in feces decreases by 25 U/g, the risk of diabetes increases by 35% [108]. The IAP activity of fecal excretion reflects the level of IAP production, digestion, and degradation in the intestine. IAP activity may be regulated by different factors, especially dietary conditions [69]. A survey found that more than 65% of healthy people suffer from “early metabolic syndrome” [71,109]. Oral IAP supplementation is a treatment for early prevention and treatment of early diabetes and/or other dominant or early metabolic diseases. Researchers believe that any treatment target should at least maintain or restore a healthy level of IAP in feces (about 65.0 U/g feces) [108]. Other treatments may involve upregulating IAP to improve immune levels, such as short chain fatty acids (such as sodium butyrate and sodium propionate), thyroid hormone, curcumin ω- 3 fatty acids, etc. [33,110,111]. In addition, eating corn oil can increase IAP secretion in rats [112]. This may be the physiological response of the body to prevent high-fat diet-related endotoxemia by secreting IAP [71]. Therefore, further study on the mechanism of IAP deficiency is of great significance for understanding the pathophysiology of T2DM. Imbalance in intestinal flora may be related to the incidence of metabolic syndrome and diabetes mellitus [113]. IAP plays two very important physiological roles in the intestinal bacterial environment: firstly, IAP can help maintain the normal structure of intestinal flora; secondly, IAP has the ability to detoxify bacterial toxins. IAP knockout mice had fewer bacteria than wild-type littermates [77]. IAP can reduce the concentration of nucleotide triphosphate, protect intestinal bacteria, and promote intestinal growth [114]. IAP can detoxify LPS toxins and destroy toxin targets by dephosphorylation (phosphohydrolysis) [115]. In addition, IAP can limit fat absorption, thereby maintaining intestinal mucosal integrity [24,116]. Oral IAP supplementation reduces intestinal sensitivity to antibiotic-induced Salmonella typhimurium and Clostridium difficile and maintains intestinal healthy homeostasis [117].

IAP deficient mice were found (akp3 gene knockout, akp3 −/−) to be diagnosed as metabolic syndrome, followed by presenting with T2DM symptoms. At the same time, because IAP can detoxify LPS and reduce metabolic endotoxemia, researchers used IAP to supplement mice orally. The results showed that oral supplement of IAP not only can prevent but also can treat metabolic syndrome and high-fat diet (HFD)-induced T2DM in wild-type mice [71]. The etiology of metabolic syndrome caused by HFD is considered to be related to metabolic endotoxemia caused by endotoxin entering the blood [62]. Other studies have found that HFD may destroy the balance of intestinal flora and lead to barrier dysfunction, after which endotoxin enters the blood through the intestinal epithelium and translocates to systemic circulation [44]. When endotoxemia occurs, IAP can be used as an effective oral supplement for preventing or treating endotoxemia, thereby protecting the host from the effects of metabolic syndrome [75]. In addition, IAP can also prevent diabetes and metabolic syndrome induced by antibiotic use in mice [118]. The protective effect of intestinal ALP has also been confirmed in transgenic mice. The overexpression of IAP in the gastrointestinal tract can reduce the HFD-induced diabetes phenotype by improving the intestinal barrier [119].

## 7. Other Diseases

ALP supplementation has a probiotic effect on multiple organs of the body, including the treatment of intestinal-, liver-, and kidney-related diseases [120,121]. ALP supplementation can detoxify a variety of proinflammatory mediators in the intestinal cavity. Among them, ALP has the most significant detoxification function on LPS (also known as endotoxin). In recent years, LPS has been proved to be one of the key mediators connecting the development of intestinal and liver diseases and many other systemic diseases. LPS consists of core polysaccharide, O-antigen, and lipid A. ALP can remove the phosphate group on LPS lipid A by dephosphorylation, so as to relieve the toxicity of LPS [122,123,124]. In a study of liver inflammation, it was found that LPS induced chronic inflammation by stimulating the expression of TLR4 in tissue cells [125,126]. When the IAP gene was knocked out, the tolerance of mouse intestinal environment to LPS decreased significantly and was more vulnerable to intestinal validation [49,71]. When intestinal injury occurs, LPS is more likely to enter the blood, causing liver injury and even serious diseases. When disease occurs, IAP supplementation—whether oral or injection—has been widely used to prevent and treat inflammatory diseases. For example, oral administration of recombinant ALP can prevent alcohol-induced hepatic steatosis and chronic liver failure [127,128]. Although IAP may be partially degraded in the stomach, oral administration after mixing IAP in drinking water is a very simple route of administration. A large number of experiments have proved that oral ALP supplementation can effectively increase the concentration of IAP in the intestinal cavity [77,86]. At the same time, after IAP supplementation in drinking water, the concentration of serum LPS in intestinal vena cava and portal vein decreased significantly, which will greatly reduce the harm of LPS translocation to the liver, so as to further protect the liver and reduce the development of malignant inflammatory circulation and liver fibrosis [129]. In human clinical studies, duodenal and enteral IAP was administered to patients with severe ulcerative colitis for 7 days, and no human safety problems, adverse events, or side effects of ALP were reported [130].

With regard to intestinal flora, it was found that the number of intestinal bacteria in IAP-ko mice decreased overall and that oral supplementation of IAP in WT mice could quickly restore the normal intestinal flora of mice affected by antibiotics [77]. IAP prevents HFD-induced metabolic endotoxemia by regulating intestinal flora [71]. In a zebrafish model, zebrafish lacking IAP are highly sensitive to LPS toxicity. IAP plays a crucial role in promoting mucosal tolerance to intestinal resident bacteria [122]. Knockout of the intestinal ALP gene (AKP3) in mice leads to metabolic abnormalities, resulting in visceral fat accumulation and hepatic steatosis [131]. On the other hand, an investigation found that the endogenous IAP level of rats decreased with age and that metabolic syndrome was common in older animals [132,133]. This suggests that “IAP deficiency” may be an inducement leading to metabolic syndrome and that the oral supplementary IAP dose can be easily adjusted to achieve the purpose of preventing or treating metabolic syndrome [77].

Studies have demonstrated that the use of ALP in the treatment of sepsis-induced AKI is promising [134,135,136]. In the treatment of acute renal injury, high-dose ALP (75 U/kg + 25 U/kg/h intravenous injection) can significantly increase serum ALP activity, making serum ALP about 5 times higher than baseline [137]. In a sheep model of sepsis, ALP can reduce inflammation and improve lung function without adverse reactions [16]. Intravenous ALP is preferentially delivered to blood and to liver, kidney, and other abdominal organs at a lower dose. Therefore, in the treatment of renal injury, a high concentration of serum ALP level is required to increase the grade of renal tissue [138]. In addition, rodent model studies of isolated renal ischemia-reperfusion injury have shown that ALP treatment can reduce renal tubular injury [139].

For women at high risk of pregnancy complications due to LPS infection, the use of supplementary AP isozymes may be an attractive treatment option [140]. In a mouse model study, IAP inhibited LPS to play an inflammatory role by upregulating the expression of autophagy-related genes (ATG5, ATG16, IRGM1, TLR4) in the mouse small intestine. Oral ALP can prevent the immune stimulation of LPS in blood to the liver. ALP can remove the phosphate group on LPS and eliminate the toxicity of LPS. At the same time, ALP can reduce TLR4, TNF-a, matured IL-1β, and NF-κB expression by upregulating the level of mir146a in mouse liver tissue, which in turn reduces the inflammatory response of the liver [141]. It was found that endogenous IAP decreased during liver fibrosis, resulting in intestinal barrier dysfunction and fibrosis deterioration. Oral IAP can protect the intestinal barrier and further prevent the development of liver fibrosis through a TLR4-mediated mechanism [131]. Although the mechanism is unclear, oral and intravenous ALP play a renal protective role in various sepsis animal models [16,129,142,143].

## 8. Prevent Aging

Recent studies have pointed out that many functions of IAP are related to aging and inflammation. IAP may protect intestinal barrier function by upregulating intestinal tight junction protein [75,76,116]. IAP can also regulate the growth of intestinal symbiotic bacteria and maintain the healthy homeostasis of intestinal microbiota [77,114]. The supplement of ALP not only has preventive and therapeutic effects on diseases but also is related to anti-aging diseases. In a mouse model, it was found that the content of ALP in mouse intestine decreased with age [78]. With an increase in age, intestinal permeability increases, accompanied by an increase in enterogenous and systemic inflammation. All these phenotypes were significantly more pronounced in IAP-deficient animals. Oral IAP supplementation can significantly reduce age-related intestinal permeability and intestinal-derived systemic inflammation, reduce weakness, and prevent aging [144]. In addition, IAP supplementation is associated with maintaining the homeostasis of the intestinal microbiota during aging [145], thereby reducing age-related intestinal permeability and intestinal-derived systemic inflammation, reducing weakness and preventing aging [86].

## 9. ALP in Raw Milk

Table 2 summarizes the content of ALP in raw milk of different animals. It can be seen from the results that among the non-human milk types, the content of alkaline phosphatase in raw sheep’s milk is generally high [146,147,148,149]. In a study of human milk, the average ALP activity of breast milk sampled in the first week after birth (6400 U/L) was 250% higher than that of breast milk sampled in the second week (2500 U/L). The highest ALP activity in breast milk in the first week was more than 20,000 U/L [10]. In addition, there are great differences in ALP activity among the same animals. In a study of dairy cows, it was found that the difference in ALP activity in the milk of individual dairy cows was as high as 40 times. At the same time, the onset of mastitis may cause a large increase in ALP activity in milk [149,150,151,152,153,154,155]. In a study of goat milk and sheep milk, it was pointed out that the activity of ALP in goat milk was affected by season and lactation stage. Studies have shown that ALP activity is higher in sheep’s milk produced in summer or in late lactation [156,157,158]. ALP activity was highest in human colostrum and then decreased with the increase in lactation time [159]. Some studies have found that ALP activity remains in heat-processed cheese [160,161].

**Table 1 foods-11-01212-t001:** Preventive and therapeutic effects of alkaline phosphatase (ALP) on diseases.

Animal	Treatment Method	Treatment Dosage	Treatment Frequency	Duration	Disease	Description	Reference
Rats	Oral	Base formula	Feed	4 days	NEC	Protect intestine	[162]
Rats	Oral	0.4, 4,or 40 U/kg	Feed	3 days	NEC	Preserving the intestinal epithelial barrier function	[163]
Rats	Oral	0.4, 4, or 40 U/kg	Feed	1 day	NEC	Decreased nitrosative stress; decreased intestinal TNF-α mRNA expression; decreased LSP translocation into the serum	[164]
Rats	Oral	0.4, 4, or 40 U/kg	Feed	1 day	NEC	Reduces systemic proinflammatory cytokine expression	[165]
Rats	Oral	700 U/kg	Gastroesophageal catheter	6 days	IBD	Protection against bacterial translocation	[30]
Rats	Oral	1035 U	Drinking water	8 days	IBD	Reduced mRNA levels for TNF-α, IL-1β, IL-6, and iNOS	[26]
Mice	Oral	200 U/kg	Gavage	4 h	Liver injury	Reducing LPS toxicity and preventing liver injury	[141]
Mice	Oral	6 U/ml	Gavage	8 days	Food allergic	Reduction in CD103+CD11b+ dendritic cells and TGF-β-producing regulatory T cells in the mesenteric lymph nodes	[8]
Mice	Oral	6 U/ml	Gavage	8 days	Food allergic	Regulation of intestinal microbial community structure	[102]
Mice	Oral	100 or 300 U	Gavage	2 weeks	Chronic colitis	Inhibit the activation of intestinal epithelial cells and peritoneal macrophages and attenuate chronic murine colitis	[166]
Mice	Oral	100 U/mL	Drinking water	21 days	Metabolic syndrome	Alterations in the composition of the gut microbiota	[118]
Mice	Oral	150 or 300 U/mL	Drinking water	48 h	Gut barrier dysfunction	Decreased expression of intestinal junctional proteins and impaired barrier function	[116]
Mice	Oral	200 U/mL	Drinking water	5 days	Bacterial infections	Protected mice from antibiotic-associated bacterial infections	[117]
Mice	Oral	100 U/mL	Drinking water	11 weeks	Metabolic syndrome	Inhibits absorption of endotoxin with dietary fat; prevents or reverses metabolic syndrome	[71]
Mice	Oral	300 U/mL	Drinking water	7 days	Inflammatory bowel diseases	Tissue myeloperoxidase activity and proinflammatory cytokines were significantly decreased	[49]
Mice	Oral	200 U/mL	Drinking water	4 days	Liver fibrosis	Protects the gut barrier and development of liver fibrosis via a TLR4-mediated mechanism	[129]
Mice	Oral	100 U/mL	Drinking water	Lifetime	Aging	Targeting crucial intestinal alterations, including gut barrier dysfunction, microbiome dysbiosis, and endotoxemia	[86]
Mice	Oral	200 U/mL	Liquid diet	10 days	Hepatosteatosis	Ameliorated the activation of hepatic stellate cells and prevented their lipogenic effect on hepatocytes	[129]
Human	Oral	30,000 U	Infusion pump	7 days	Ulcerative colitis	Decrease the C-reactive protein and stool calprotectin levels	[130]
Rats	Injection	1000 U/kg	Intraperitoneal	4 days	Acute liver failure	Reduced LPS activity and hepatic TLR4 expression	[128]
Rats	Injection	1000 U/kg	Intraperitoneal	24 h	Acute kidney injury	Reduce renal inflammation; dephosphorylation of ATP and LPS	[142]
Rats	Injection	500 U/kg	Intraperitoneal	5 min	Partial liver resection	Attenuate both hepatic and pulmonary injury	[167]
Mice	Injection	100 U/mL	Intestinal loop	2 h	Intestinal flora disorder	Inhibiting the concentration of luminal nucleotide triphosphates	[114]
Mice	Injection	15 U/mL	Intravenous	1 day	Sepsis	Normalize body temperature	[168]
Mice	Injection	150 U	Intravenous	5 min	Pregnancy complications	Protects early pregnancy defects	[140]
Mice	Injection	150 U/kg	Intravenous	72 h	Secondary peritonitis	Attenuates the inflammatory response both locally and systemically and reduces associated liver and lung damage	[169]
Human	Injection	5.6 U/kg/h	Intravenous	36 h	Cardiac surgery	Endogenous alkaline phosphatase release	[170]
Human	Injection	67.5 U/kg + 132.5 U/kg/24 h	Intravenous	48 h	Acute kidney injury	Reductions in the systemic markers C-reactive protein, IL-6, and LPS-binding protein and in the urinary excretion of kidney injury molecule-1 and IL-18	[135]
Piglet	Injection	1, 5, or 25 U/kg/h	Intravenous	4 h	Acute kidney injury	Increased serum or renal tissue AP activity	[137]
Piglet	Injection	1, 5, or 25 U/kg/h	Intravenous	4 h	Cardiac surgery	Increased kidney and liver tissue alkaline phosphatase activity	[171]

**Table 2 foods-11-01212-t002:** Content of alkaline phosphatase in raw milk.

Source	Content (U/L)	Reference
Human	74.10–20,000.00	[10,159]
Cow	5.29–1155.00	[146,147,148,149,155,172,173,174,175,176]
Buffalo	15.05–117.21	[152,154]
Goat	2.28–1786.00	[147,148,149,150,156,157,172,173,175]
Sheep	722.00–2814.00	[147,148,149,150,156,175]
Equine	3.12–20.81	[174]
Donkey	35.04–37.06	[177]
Camelids	12.70–94.14	[153,175,178,179]

## 10. Conclusions

This paper summarizes the immunoprotective effect of ALP on a host, especially infants. Whether administered orally or by injection, ALP will have preventive or therapeutic effects on many host diseases, including enteritis, diabetes, liver diseases, and kidney diseases. At the same time, the activity of ALP in raw milk of different animals was summarized. At present, many studies have verified the prebiotic and medicinal effects of ALP. ALP can be synthesized by animals themselves, and ALP ingested from external sources also has a variety of prebiotic effects to protect the health of animal hosts. In the future, research of infant food production and additives on the treatment and prevention of infant diseases will receive more extensive attention. ALP supplementation has a positive effect on the healthy growth of infants, human health, and the extension of life span. It is hoped that through this review, consumers and producers will pay more attention to ALP. Especially for infants with incomplete immune development, ALP supplementation is conducive to healthy growth of infants.

## Data Availability

Not applicable.

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
