# Peer review of "Protective Effect of Alkaline Phosphatase Supplementation on Infant Health"

_foods, 2022, doi:10.3390/foods11091212_

Round 1
Reviewer 1 Report
The review of Wu et al. is highlighting the importance of alkaline phosphatase on infant health. The review is nicely written and information are adequately summarized. However there are minor issues that should be corrected. Line 65: Put "." at the end of sentence. Line 86: Bi directionally should be "bi-directionally". Bacterial names should be italicized throughout the manuscript. .Author Response
The review of Wu et al. is highlighting the importance of alkaline phosphatase on infant health. The review is nicely written and information are adequately summarized. However there are minor issues that should be corrected. Line 65: Put "." at the end of sentence. Line 86: Bi directionally should be "bi-directionally". Bacterial names should be italicized throughout the manuscript. .
Response: Thank you very much for your comments. We have revised the manuscript according to your suggestion
Reviewer 2 Report
- The title is vague. It is hard for the readers to understand from the title whether the review is about endogenous ALP or supplemented ALP?
- A section dedicated to literature review criteria including inclusion and exclusion criteria needs to be added.
- Although the manuscript is about the effects of ALP on infant health, several papers have been included which are not at all related to infants. Either replace these papers with appropriate ones, or connect the cited papers properly with the theme if the manuscript.
- Try to avoid using the word ‘dysbiosis’ since the definition of good or bad bacteria is relative.
- The manuscript needs discussion of how indigenous ALP protects against mucosal inflammation. Also, discussion regarding how dietary factors like phytochemicals can provide health beneficial effects by modulating mucosal ALP would be interesting to the readers.
- Rows in the ‘Action mechanism’ column in table 1 does not match with the remaining rows. Please make corrections.
- The conclusion is loosely written. Critical discussion of future prospects is lacking.
Reviewer 3 Report
Dear Authors,
Thank you for the opportunity to review this paper. I my opinion, the manuscript describes a very relevant and interesting topic the immunoprotected effect of alkaline phosphatase on host, especially infants.
In my opinion, manuscript requires major corrections, especially in terms of methodology description, publication selection protocol (where, in what databases, what keywords, etc.).
Comments and Suggestions for Authors:
1.Keywords should be different than the words in the title.
2. I have great doubts about the lack of a systematic and clear methodological approach to collecting and selecting the base literature for analysis.
The major advantage of systematic reviews is that they are based on the findings of comprehensive and systematic literature searches in all available resources, with minimization of selection bias avoiding subjective selection bias. It is important to use and describe an unambiguous methodological (systematic) approach to the selection of publications for review.
Therefore, can the authors consider supplementing the work with an appropriate methodological chapter? A chapter in which they will show the protocol of the procedure in the selection of source literature (eg PRISMA). The lack of such indications, despite the fact that the article is clearly expert, may suggest a bias and some orientation in the selection of source materials.
Thank you.
Round 2
Reviewer 3 Report
Dear Authors,
Thank you for the opportunity to review this paper again.
The text of the publication has been slightly improved. My main comments regarding the subjectivity of the review were not taken into account by the authors.
I acknowledge the authors' opinion. I have no substantive comments to the reviewed manuscript, so I leave the final decision to the Editor.
Thank you.
Author Response
Thank you very much for your careful and meticulous suggestions, which have benefited a lot.